# Evaluating the modulation of peripheral immune profile in people living with HIV and (Neuro)cysticercosis

Yakobo Leonard Lema[1]*, Ulrich Fabien Prodjinotho[2,3], Charles Makasi[1,4], Mary-Winnie A. Nanyaro[1], Andrew Martin Kilale[1], Sayoki Mfinanga[1,5,6], Dominik Stelzle[3,7], Veronika Schmidt[3,7], Hélène Carabin[8,9,10,11], Andrea Sylvia Winkler[3,7,12,13], Eligius F. Lyamuya[14], Bernard J. Ngowi[1,15], Mkunde Chachage[15,16], Clarissa Prazeres da Costa[2,3,17]*

1 Muhimbili Medical Research Center, National Institute for Medical Research (NIMR), Dar es Salaam, Tanzania, 2 Institute for Medical Microbiology, Immunology, and Hygiene, Technical University of Munich (TUM), Munich, Germany, 3 Center for Global Health, Technical University of Munich, Munich, Germany, 4 Kilimanjaro Christian Medical University College, Moshi, Tanzania, 5 Kampala International University, Kampala, Tanzania, 6 School of Public Health, Muhimbili University of Health & Allied Sciences (MUHAS), Dar Es Salaam, Tanzania, 7 Department of Neurology, School of Medicine and Health, Technical University Munich (TUM), Munich, Germany, 8 Faculty of Veterinary Medicine, University of Montreal, Saint-Hyacinthe, Quebec, Canada, 9 School of Public Health, University of Montreal, Montreal, Quebec, Canada, 10 Research Group on Epidemiology of Zoonoses and Public Health (GREZOSP), Saint-Hyacinthe, Quebec, Canada, 11 Public Health Research Center of the University of Montreal and the CIUSSS of Center-Sud-de-l'île-de-Montréal (CReSP), Montreal, Quebec, Canada, 12 Department of Community Medicine and Global Health, Institute of Health and Society, University of Oslo, Oslo, Norway, 13 Department of Global Health and Social Medicine, Harvard Medical School, Boston, Massachusetts, United States of America, 14 Department of Microbiology & Immunology, Muhimbili University of Health & Allied Sciences (MUHAS), Dar Es Salaam, Tanzania, 15 Mbeya College of Health & Allied Sciences, University of Dar Es Salaam, Mbeya Tanzania, 16 Mbeya Medical Research Center, National Institute for Medical Research (NIMR), Mbeya, Tanzania, 17 German Center for Infection Research (DZIF), partner site Munich, Germany

* yakobolema@gmail.com (YLL); clarissa.dacosta@tum.de (CPdC)

**Data Availability Statement:** All relevant data are in the manuscript and its supporting information files.

## Abstract

### Background

The parasitic infection caused by *Taenia solium* represents a significant public health concern in developing countries. Larval invasion of body tissues leads to cysticercosis (CC), while central nervous system (CNS) involvement results in neurocysticercosis (NCC). Both conditions exhibit diverse clinical manifestations, and the potential impact of concomitant HIV infection especially prevalent in sub-Saharan Africa on peripheral and CNS immune responses remains poorly understood. This study aimed to identify the potential impact of HIV coinfection in CC and NCC patients.

### Methodology

A nested study within a cross-sectional analysis in two Tanzanian regions was performed and 234 participants (110 HIV+ and 124 HIV-) were tested for cysticercosis antibodies, antigens, CD4 counts and serum Th1 and Th2 cytokines via multiplex bead-based immunoassay. 127 cysticercosis seropositive individuals underwent cranial computed tomography

**Funding:** BJN was supported by a grant from the German Federal Ministry of Education and Research (Bundesministerium für Bildung und Forschung, BMBF) [81203604] under the "Research Networks for Health Innovations in Sub-Saharan Africa" (RHISSA) initiative. The funders had no role in study design, data collection and analysis, decision to publish, or preparation of the manuscript. More information about the funder can be found at https://www.cystinet-africa.net/funder/.

**Competing interests:** The authors have declared that no competing interests exist.

(CCT) and clinical symptoms were assessed. Multiple regression analyses were performed to identify factors associated with cytokine modulation due to HIV in CC and NCC patients.

## Results

Serologically, 18.8% tested positive for cysticercosis antibodies, with no significant difference HIV+ and HIV+. A significantly higher rate of cysticercosis antigen positivity was found in HIV+ individuals (43.6%) compared to HIV- (28.2%) (p = 0.016). CCT scans revealed that overall 10.3% had active brain cysts (NCC+). Our study found no significant changes in the overall cytokine profiles between HIV+ and HIV- participants coinfected CC and NCC, except for IL-5 which was elevated in HIV+ individuals with cysticercosis. Furthermore, HIV infection in general was associated with increased levels of pro-and some anti-inflammatory cytokines e.g. TNF-α, IL-8, and IFN-γ. However, based on the interaction analyses, no cytokine changes were observed due to HIV in CC or NCC patients.

## Conclusions

In conclusion, while HIV infection itself significantly modulates levels of key cytokines such as TNF-α, IL-8, and IFN-γ, it does not modulate any cytokine changes due to CC or NCC. This underscores the dominant influence of HIV on the immune system and highlights the importance of effective antiretroviral therapy in managing immune responses in individuals coinfected with HIV and CC/NCC.

### Author summary

Our study evaluates the interplay of immune responses in individuals coinfected with HIV and neurocysticercosis (NCC) in resource-limited settings. We analyzed cytokine profiles among 234 participants, discovering that HIV infection significantly modulates various key cytokines such as TNF-α, IL-8, and IFN-γ. Notably, our results indicate that while HIV has a dominant influence on cytokine levels, it does not cause additional cytokine alterations specifically due to NCC. This suggests that the immunomodulatory effects of NCC are minimal in the presence of HIV, pointing to the overarching impact of HIV on the immune system. Our findings emphasize the complexity of immune responses in coinfected individuals and underscore the critical role of effective antiretroviral therapy. Insights from our study are crucial for refining therapeutic strategies in managing such complex coinfections in endemic regions.

## Introduction

Cysticercosis, resulting from the ingestion of *Taenia solium* eggs, poses a significant public health challenge, particularly in low- and middle-income countries [1]. This parasitic infection leads to the development of cysticerci in various body tissues, including muscles, eyes, and heart. When these larvae invade the central nervous system, the condition is referred to as neurocysticercosis (NCC) This parasitic infection presents a broad spectrum of clinical manifestations, ranging from asymptomatic cases to severe neurological conditions, including seizures, neurological deficits, and hydrocephalus [2–5]. Notably, viable cysts without significant imaging changes are typically asymptomatic. However, the onset of seizures marks a critical

transition from asymptomatic to symptomatic NCC. The presence of asymptomatic NCC within communities poses significant challenges. At the population level, it hinders the detection of tapeworm carriers since the identification of symptomatic cases typically prompts targeted screening efforts to interrupt transmission. Meanwhile, at the individual level, there is a high risk to develop severe symptoms upon anthelminthic therapy mass drug administration (MDA) to treat other (intestinal) helminths [6,7].

Diagnosing NCC accurately is a challenge due to its varied presentations and the limitations of diagnostic tools such as ELISA, with sensitivity varying across different disease stages and cyst characteristics [8–11]. Despite the high specificity of the antibody-ELISA test for *T. solium*, confirming exposure, its inability to differentiate between active and inactive infections highlights the need for more definitive diagnostic approaches [12]. While most Antigen-ELISA tests are effective in detecting active infection and for monitoring cyst clearance post treatment, they perform worse in detecting calcified and single lesions [6,12].

Neurocysticercosis is the leading cause of acquired epilepsy and a significant contributor to late-onset epilepsy in tropical regions around the world [13,14]. Experts estimate that in areas endemic for *T. solium*, large proportions of people with acquired epilepsy show lesions consistent with NCC [3,15,16]. In southern Tanzania, a hospital-based study even found prevalence proportions of 31 to 38% for NCC among people with epileptic seizures [11]. Other clinical manifestations of NCC include acute and chronic headaches, signs or symptoms of intracranial hypertension, neuropsychiatric disorders, and focal neurological deficits [17]. These clinical symptoms result mainly from the host's immune response orchestrated by cytokines released in response to the presence of cysts and, to a much lesser extent, to worm antigens [15].

When cysticerci infect the brain, the resulting pro-inflammatory response leads to disruption of the blood-brain barrier and infiltration of $CD4^+$ and $CD8^+$ T cells, B cells, macrophages, and dendritic cells at the site of infection following up-regulation of cell adhesion molecules such as ICAM-1 and VCAM-1 [18,19]. These cells produce cytokines that orchestrate the immune responses, leading to the death of the invading parasite. Research conducted in murine models, pigs, and humans has discovered mixed Th1/Th2 cytokine production in response to the presence of the cyst [20–22]. This immune response initially predominates as Th1 inflammatory responses (e.g., TNF-α, IFN-γ, IL-12) during the infection's initial phase, but as the disease progresses, it shifts to a permissive Th2 response (IL-4, IL-5, IL-13) [23]. Also, it is important to note that individuals without infection of the CNS have a balanced Th1/Th2 immune profile. However, once the cyst infects the CNS, even asymptomatic individuals show a predominantly Th2-skewed response, characterized by elevated levels of IL-4, IL-5, IL-13, and IgG [24]. In contrast, symptomatic patients exhibit increased levels of TNF-α, IL-17, IL-23, and sICAM-1, indicating a different immune response [23]. These immunological parameters and clinical presentations might be further disturbed by the number of cysts in the brain, their location, and, importantly, in coinfection settings with other chronic diseases such as HIV [25]. Indeed, many areas affected by *T. solium* in sub-Saharan Africa (SSA) are co-endemic to HIV, and a recent WHO fact sheet on HIV and AIDS reports that almost 25.5 million people in Sub-Saharan Africa (SSA) are living with HIV/acquired immunodeficiency syndrome (AIDS) [26]. While studies have demonstrated HIV interactions with tuberculosis, malaria, filarial infections, and some soil-transmitted helminths, researchers know less about the coinfection of HIV and neuro(cysticercosis) [27–32]. HIV-related immune activation has been associated with an increased risk of NCC due to inflammation of the CNS and weakened immunity with further clinical manifestations, including neurological complications [33–36]. However, whether and how the host immune responses, especially the cytokine responses, are impaired in HIV and NCC settings, driving such complications is still not known. Therefore,

this study aimed to investigate the difference in peripheral immune responses to NCC between HIV+ and HIV- participants.

## Methods

### Ethics statement

Ethical approval for this study was obtained from the Directorate of Research and Publications, Muhimbili University of Health & Allied Sciences (MUHAS), Dar es Salaam (Ref. No. DA.282/298/01. C/). The proposal for the Cystinet Africa project received ethical clearance from the National Institute for Medical Research (NIMR) with Ref. NIMR/HQ/R.8a/Vol. IX/ 2529, also the Technical University of Munich, School of Medicine and Health, Ethics Committee (Ref: 537/18) and Ref: 215/18S).

All participants were informed about the potential risks and benefits of all diagnostic tests, including CT scan, for those eligible participants. The study did not include pregnant women and children under 14 years of age. Written informed consent was sought from all study participants, parents, or guardians of a minor, and assent from children between 14 and 17. All participants who could not read and write were asked to have a relative who could read and write as their witness. For all control participants recruited, an HIV test was required, and before each test, specially trained nurses were tasked with counseling the potential participants. Participants were informed of their HIV test results, and all positive cases from the community were referred to the district hospital CTC for care and management.

### Study design and recruitment of study participants

This investigation constitutes a nested cross-sectional immunological study within a broader paired cross-sectional framework conducted in the endemic regions of Chunya district (Mbeya region) and Iringa rural district (Iringa region), Southern Highlands of Tanzania, between June 2018 and March 2021[37]. The primary aim was to explore the immunological profiles in HIV+ individuals with *T. solium* cysticercosis.

The larger paired cross-sectional study gathered information on the seroprevalence of cysticercosis, assessed risk factors associated with cysticercosis transmission, and evaluated knowledge, perceptions, and practices related to cysticercosis among 1291 HIV+ individuals, randomly recruited from Care and Treatment Centers (CTCs) and their matched controls from the community. The controls were HIV-ve, age- (within ±5 years), and sex-matched and residing within 100 meters of the index case household to control for potential variations in exposure to *T. solium*.

All participants underwent a clinical evaluation and provided a serum sample for serological testing. This provided a foundation for the nested study to further investigate immunological responses.

Participants positive for cysticercosis, along with their matched counterparts and a randomly selected group were invited to undergo cerebral computed tomography (CT) scans for NCC lesion assessment and for serum cytokine analysis for assessment of the peripheral immune responses. In total 234 participants underwent the immunological assessment, 167 also received CT scans enabling a comprehensive evaluation of the interplay between *T. solium* infection, HIV status, and immune responses.

### Sample collection and testing

Each participant provided 15 ml of blood by venipuncture. Two ml of serum were collected and stored at -20˚C for cysticercosis antibody and antigen testing. Ten ml of blood, collected

in an EDTA tube, underwent analysis for the CD4 count test. Technicians centrifuged the remaining volume to obtain plasma for HIV viral load testing. HIV- participants from the same community (endemic controls) were given HIV counseling during recruitment. Those who tested positive for HIV were referred to the CTC for further treatment, while those who tested negative provided 5 ml of blood, which was further separated to obtain serum for cysticercosis antigen and antibody detection. The study utilized the LDBio cysticercosis Western Blot IgG test from LDBio Diagnostics SARL, Lyon, France, to detect circulating cysticercosis-specific antibodies targeting a recombinant antigen of *T. solium*. For antigen detection, indicating the presence of viable cysts, the study employed a monoclonal antibody-based B158C11A10 ELISA (Cysticercosis Ag-ELISA) from apDia bvba, Turnhout, Belgium [38]. The study invited all participants who tested positive for cysticercosis antibodies, antigens, or both, along with their matched partners, to Mbeya Zonal Referral Hospital for a brain CT scan and to provide blood samples for measuring serum cytokine levels using a multiplex bead-based assay.

## Cytokine detection (multiplex bead-based immunoassay)

The study analyzed cytokines associated with Th1/Th2 immune responses, focusing on pro-inflammatory and Th1 cytokines (TNF-$\alpha$, IFN-$\gamma$, IL-1$\beta$, IL-6, IL-8, IL-12, IL-17, and IL-18), Th2 cytokines (IL-4 and IL-5) anti-inflammatory cytokines (IL-10 and IL-13), and cell adhesion molecules (VCAM-1, ICAM-1). Biotechne R&D Systems, Minneapolis, MN, USA, provided custom multianalyte kits for the study's analysis, conducted using a Luminex MAGPIX xMAP 100 instrument from Luminex Corporation, Austin, TX. The team used xPONENT software (v4.3) to calculate the standard curve and individual well concentrations, applying a five-parameter regression formula.

## Characterization of NCC cases

Cranial CT scans performed by radiologic technologists confirmed NCC diagnoses. The parasites were staged based on CT scan findings: vesicular (transparent membranes and vesicular fluid), colloidal (degenerating parasites with cloudy fluid, indicative of an intense inflammatory reaction), and calcified (appearing as mineralized granulomas). Patients were categorized as follows: those with vesicular or colloidal stages were identified as having active NCC (n = 10), those with only the calcified stage were classified as having inactive NCC (n = 21), and those with mixed stages were also considered as having active NCC (n = 7).

## Statistical analysis

Data management and statistical analysis in this study were conducted through various specialized software. Initial data cleaning and processing utilized Microsoft Excel (Version 2022, Microsoft Corporation, Redmond, WA, USA). This stage was followed by univariate analyses performed using IBM SPSS Statistics for Windows (Version 26.0, Armonk, NY: IBM Corp). The distribution of variables within demographic tables was compared using the Chi-square test for bivariate analysis.

The immune response assessment involved evaluating the concentrations of various cytokines. These concentrations and their corresponding statistical analyses were plotted using GraphPad Prism software (Version 9.1.0, GraphPad Software, San Diego, California, USA). Differences in cytokine concentrations between HIV+ and HIV- groups were analyzed using the Mann-Whitney U test due to non-parametric data point distribution and stratified according to cysticercosis and neurocysticercosis status. All p-values obtained were Holm-Bonferroni

corrected to minimize Type 1 error from multiple hypothesis testing. P-values <0.05 after correction were considered statistically significant.

For reporting the data, median values along with the Standard Error of the Mean (SEM) were provided to better represent the variability and reliability of the measurements. Multivariable linear regression models were conducted to assess potential confounding factors for the effect of HIV status on cytokine levels, independent of CC status.

Detailed data supporting the figures, including cytokine concentrations, statistical analyses, and additional comparisons, are provided in S1 Table.

## Results

### Baseline characteristics of the study population

Our study included 234 participants, 110 people living with HIV and 124 HIV negative individuals, from Iringa (28.6%, n = 67) and Mbeya (71.4%, n = 167) regions. The median age was similar between HIV+ (43 years) and HIV- (42 years) participants, with no statistically significant difference in gender distribution (p = 0.73). A significant demographic difference was found in marital status: a higher proportion of HIV+ participants were unmarried (divorced or widowed) (24.6% vs. 12.1% in HIV- participants), and a smaller proportion were single (3.6% vs. 13.7% in HIV- participants) (p = 0.002).

Most HIV+ participants (72.7%) had an undetectable viral load, indicating effective adherence to antiretroviral therapy, with approximately 30% having a CD4 count below 200 cells/μL and only seven (6.4%) classified as WHO clinical stage 4. All HIV+ were on anti-retroviral treatment (ART). Further demographic and clinical characteristics are provided in S1 Text.

### Serological and radiological investigations

Building on the baseline data, we further analyzed serological and radiological indicators of cysticercosis for differences between HIV+ and HIV- participants as detailed in Table 1. Antibody tests for cysticercosis were positive in 44 (18.8%) participants, with no significant difference between HIV+ (20.9%) and HIV- (16.9%) participants (p = 0.458). Our findings demonstrate a higher proportion of HIV+ participants positive for cysticercosis antigens compared to their HIV- counterparts (43% vs. 28.2%, p = 0.016). A total of 183 participants had a cranial CT scan. Of these, 38 participants harbored brain cysts, of which 19 (50%) were

**Table 1. Laboratory and radiological findings stratified by HIV status of study participants.**

| Variables | Total n (%) | HIV Positive n (%) | HIV Negative n (%) | *P*-value* |
|---|---|---|---|---|
| **Cysticercosis Ab (234)** | | | | |
| Positive | 44 (18.8) | 23 (20.9) | 21 (16.9) | 0.458 |
| Negative | 190 (81.2) | 87 (79.1) | 103 (83.1) | |
| **Cysticercosis Ag (234)** | | | | |
| Positive | 83 (35.5) | 48 (43.6) | 35 (28.2) | 0.016 |
| Negative | 151 (64.5) | 62 (56.4) | 89 (71.8) | |
| **CT scan findings (183) ‡** | | | | |
| Active | 19 (10.3) | 7 (6.9) | 12 (14.6) | 0.204 |
| Inactive | 19 (10.3) | 12 (11.8) | 7 (8.5) | |
| No cysts seen. | 145 (79.2) | 82 (81.1) | 63 (76.8) | |

*P-values were calculated using the chi-square test, p<0.05 is considered significant.

‡ = Percentages may not total 100 due to rounding.

categorized as active NCC. Notably, among 83 participants positive for *T. solium* antigens, only 60 were also available to undergo a cranial CT scan. These scans showed that 22 participants (37%) had brain cysts at various stages: active (9), inactive (8), and mixed stage (5) while 38 (63%) showed no cysts in the brain.

To conclude, a higher proportion of HIV+ participants tested positive for cysticercosis antigens than their HIV- counterparts, underscoring the potential influence of HIV on cysticercosis progression.

## Comparison of cytokine profiles between HIV+ and HIV- participants

To explore the interaction between HIV and cysticercosis and the potential mutual influence of each infection on the immunological profile, we evaluated serum concentrations of proinflammatory and anti-inflammatory across four distinct groups: HIV+CC+ (infected with HIV and cysticercosis), HIV+CC- (infected with HIV only), HIV-CC+ (infected with cysticercosis only), and HIV-CC- (uninfected controls). The median concentrations ± SEM for each cytokine are reported. Comparable levels of all cytokines were observed between the HIV+CC + and HIV+CC- groups indicating no further impact of cysticercosis on cytokine levels in general (Fig 1). Interestingly, when comparing the levels between the HIV-CC+ and HIV-CC-

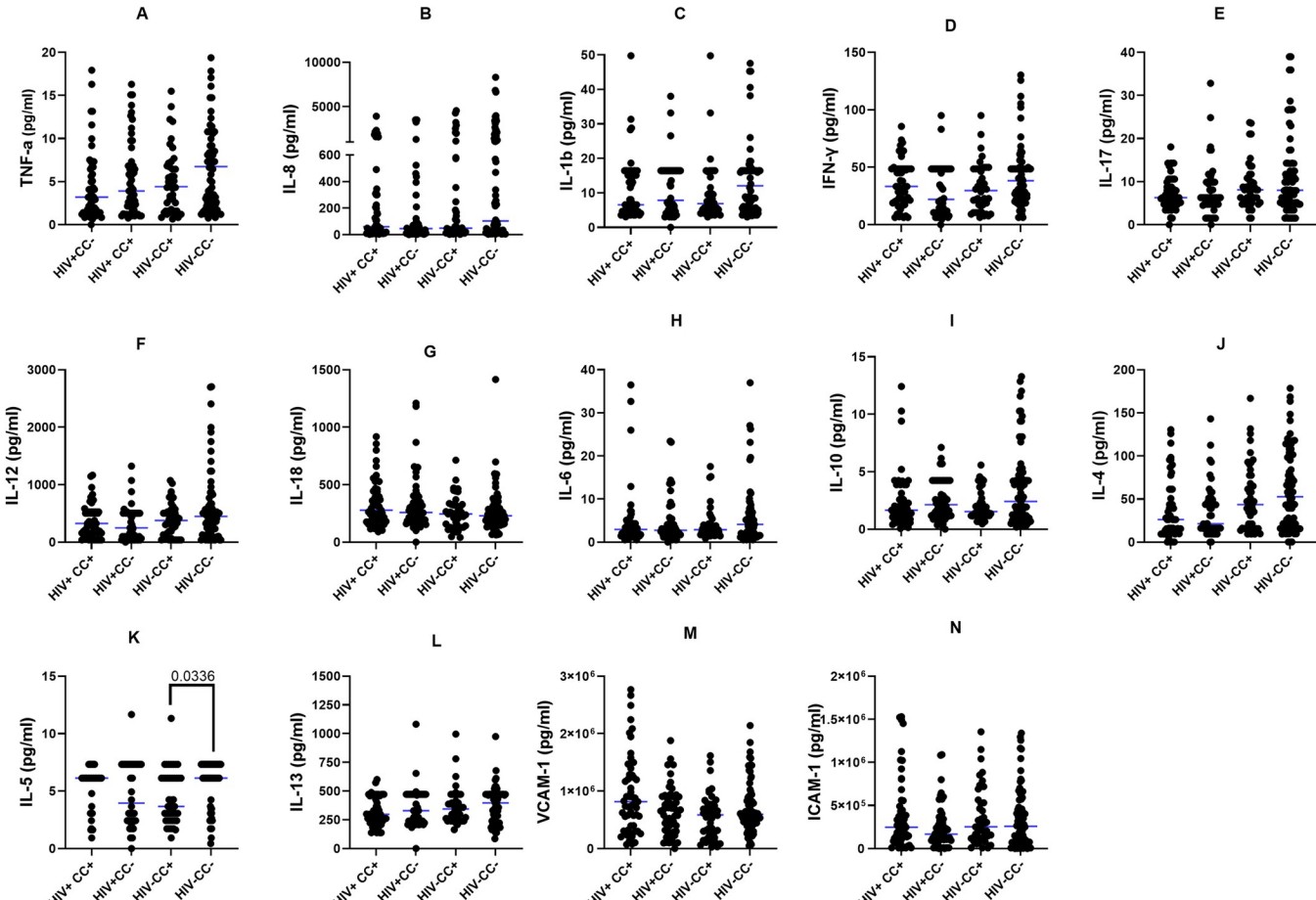

**Fig 1. The median concentrations of pro and anti-inflammatory cytokines in the sera of HIV+CC+ (n = 55), HIV+CC- (n = 55), HIV-CC+ (n = 44), and HIV-CC- (n = 80) participants, expressed as median (IQR).** Differences were analyzed using the Mann-Whitney U test, with p<0.05 indicating statistical significance.

groups, significant differences were found only in IL-5 (3.673 ± 0.414 pg/mL for HIV-CC+ vs. 6.132 ± 0.099 pg/mL for HIV-CC-, p = 0.0336) (Fig 1K) which was higher in the HIV-CC- group. All other cytokines remained comparable.

## Modulation of the cytokine profile during NCC and HIV coinfection

Expanding on our findings from stratified HIV and cysticercosis analyses, we then assessed cytokine variations across participants distinguished by HIV and NCC status taking the CT scan results into consideration as the serology is not indicative of active infection or the presence of brain cysts in general. Interestingly this further stratification resulted in no significant differences in cytokine levels across the groups. Though not significant, overall median levels in many cytokines were yet again higher in the HIV-NCC- group when compared to any of the other three infection group (Fig 2).

## Relationship between brain cyst activity and the peripheral cytokine profile in NCC

To understand how the viability of brain cysts is related to the peripheral immune response and could thereby either contribute to the development of symptomatic NCC or be indicative

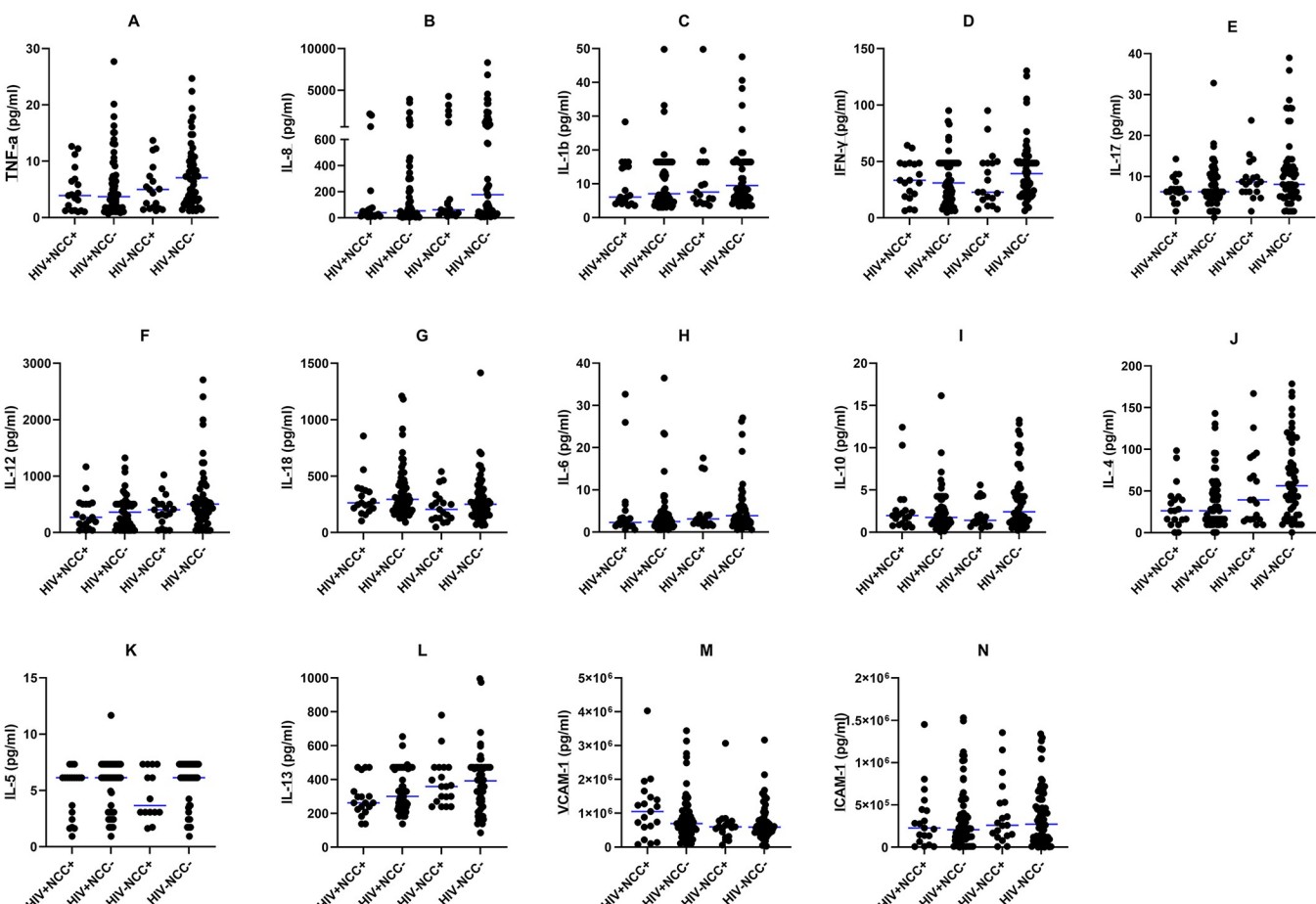

**Fig 2. The median concentration of the pro and anti-inflammatory cytokines in the sera of HIV+ NCC+ (n = 19), HIV+NCC- (n = 65), HIV-NCC- (n 64), and HIV-NCC+ (n = 19) participants, expressed as median (IQR).** Differences were analyzed using the Mann-Whitney U test, with p<0.05 indicating statistical significance.

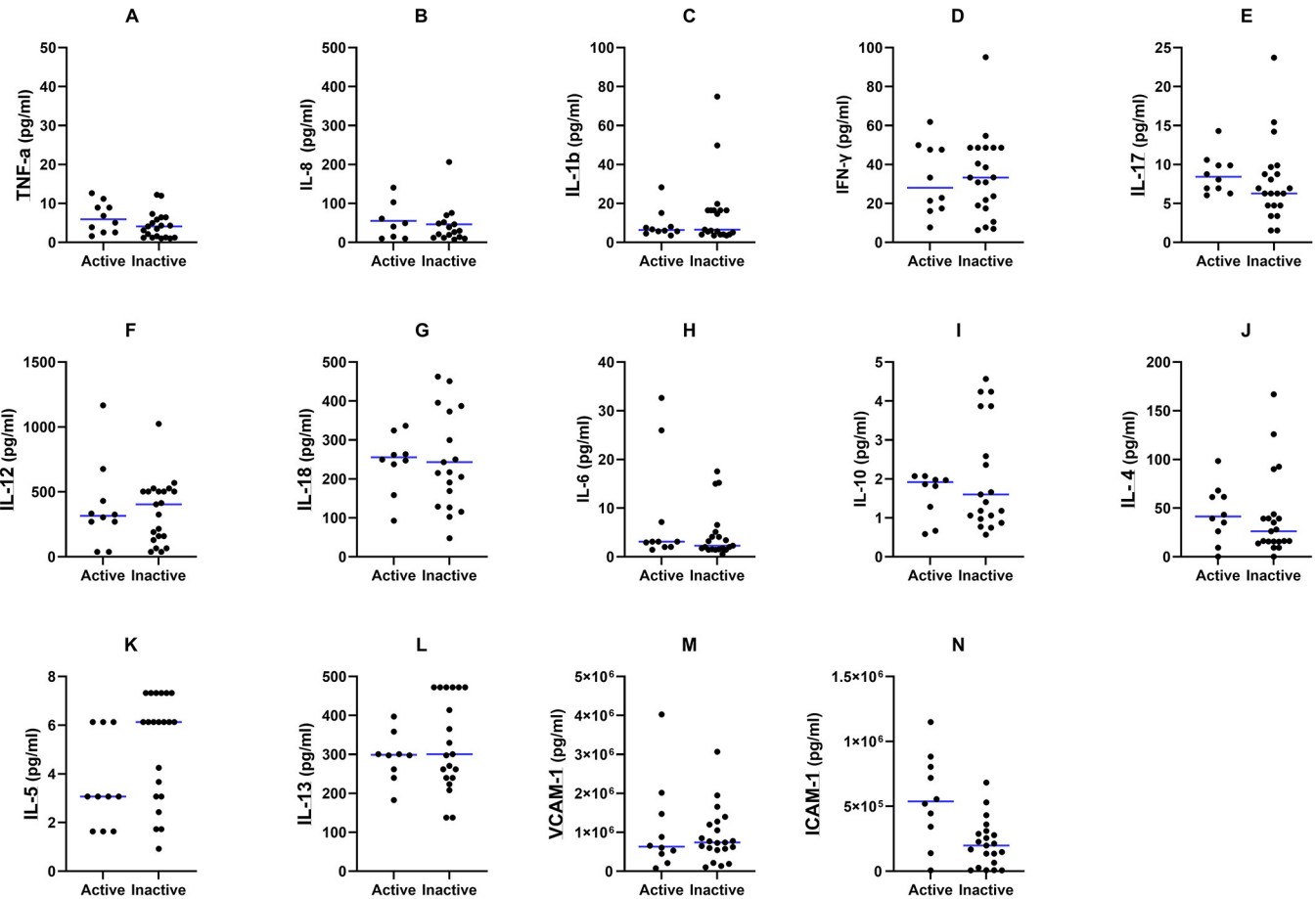

**Fig 3. The median concentrations of pro and anti-inflammatory cytokines in the sera of NCC+ patients with active NCC (n = 10) and inactive NCC (n = 21) participants, expressed as median (IQR).** Differences were analyzed using the Mann-Whitney U test, with p<0.05 indicating statistical significance.

of cyst activity (active vs. inactive) [15], we next compared cytokine levels between patients with radiologically active and inactive cysts regardless of the HIV status (Fig 3). Yet again, we observed no differences in median cytokine concentrations between the two groups. However, there was a tendency towards increased ICAM-1 levels in the group with radiologically active NCC (median 538,615 ± 95,383 pg/mL for active vs. 197,836 ± 65,315 pg/mL for inactive, adjusted p = 0.0826), suggesting a possible association with cyst activity.

Next, we evaluated the added effect of HIV in the latter analysis by stratifying the cytokine levels by HIV status for both active and inactive cyst groups. We observed no statistically significant differences in the median concentrations of all cytokines between the four groups (Fig 4).

### Assessment of risk factors associated with cytokine levels (age, sex, HIV and CC infection status)

To evaluate the effects of confounding variables on cytokine modulation (age, sex, cysticercosis status, and HIV infection), we conducted a multivariable regression analysis. Our analysis revealed significant associations for several cytokine levels to be confounded by HIV status, sometimes also by gender. Cysticercosis status, however, was not statistically associated with cytokine levels; also, we did not consistently find effect modification of HIV and CC across multiple cytokine levels.

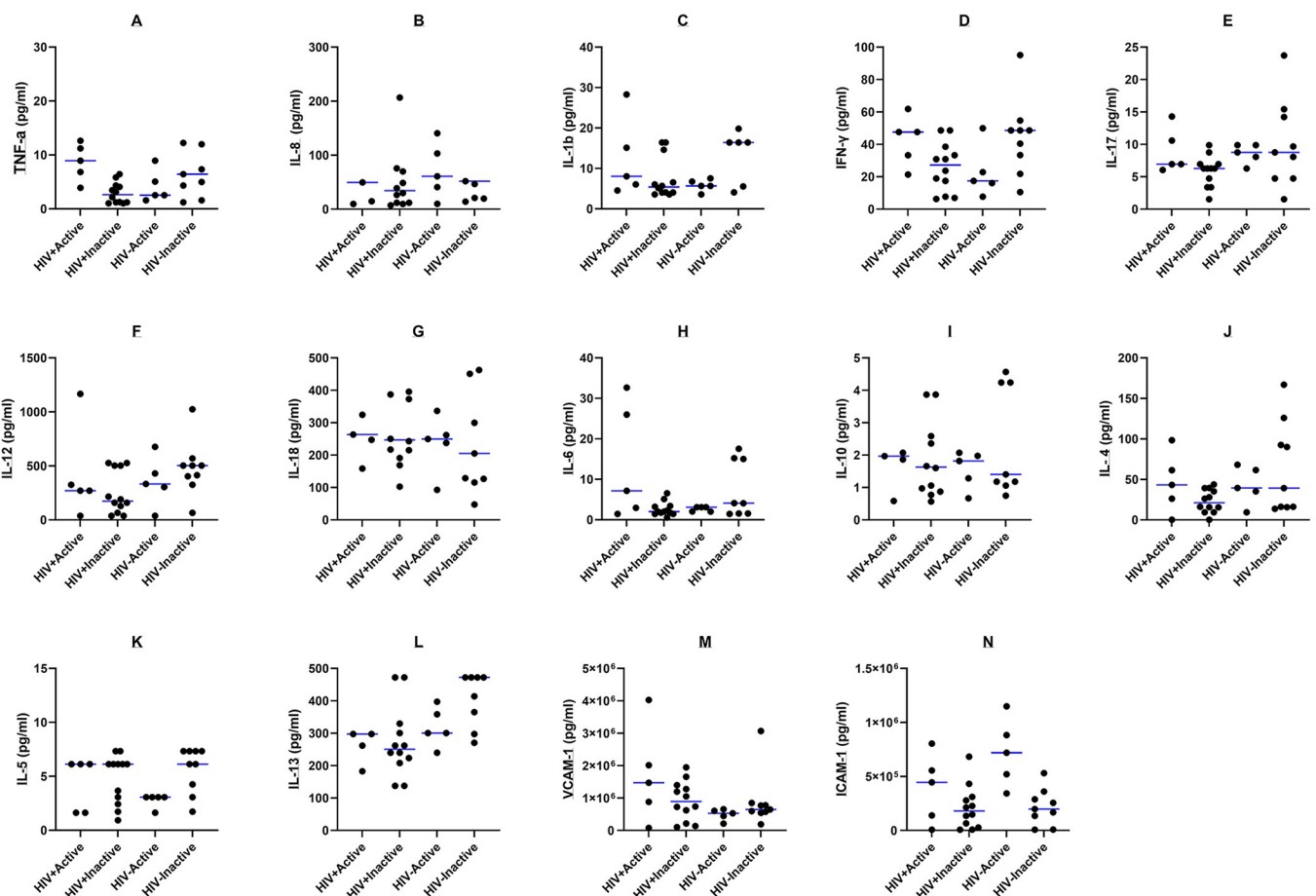

**Fig 4. The median concentrations of cytokines in the sera of both HIV+ (n = 5 active NCC, n = 12 inactive NCC) and HIV- (n = 5 active NCC, n = 9 inactive NCC) participants.** Results are expressed as median (IQR). Significant statistical differences were analyzed using the Mann-Whitney U test, with P<0.05 shown.

Our findings showed no significant effect of age on the expression levels of all cytokines measured. Gender differences significantly influenced the cytokine expression profile, male sex significantly associated with decreased levels of IL-8 (β = -0.737, p = 0.016), IL-17 (β = -0.279, p = 0.012), IL-4 (β = -0.483, p = 0.007) and ICAM-1 (β = -0.535, p = 0.013) indicating a suppressive effect of male sex on these cytokines. For the rest of cytokines measured there was no significant gender-based modulation.

Positive cysticercosis status was significantly associated with a decrease in IL-10 levels (β = -0.419, p = 0.022), while IL-5 levels were significantly increased (β = 0.229, p = 0.035).

Regarding the effect of HIV status on cytokine expression, being HIV positive was associated with significant increase in levels of TNF-α (β = 0.614, p = 0.001), IL-8 (β = 1.074, p = 0.006), IFN-γ (β = 0.441, p = 0.001), IL-17 (β = 0.301, p = 0.032), IL-12 (β = 0.539, p = 0.006), IL-6 (β = 0.688, p = 0.009) and IL-5 levels (β = 0.254, p = 0.012), all indicating significant stimulatory effects of HIV infection on these cytokines. In contrast, IL-18 levels were significantly suppressed in the presence of HIV (β = -0.262, p = 0.014). HIV status did not show a significant effect on IL-4, though a trend towards an increase was noted (β = 0.439, p = 0.053).

The interaction between cysticercosis status and HIV infection significantly decreased TNF-α (β = -0.574, p = 0.035) and IFN-γ levels (β = -0.577, p = 0.004), indicating that the coinfection may mitigate the increase typically observed with HIV alone. Similarly, IL-5 (β = -0.5,

p = 0.001) and VCAM-1 levels (β = -0.603, p = 0.009) were significantly decreased by the interaction, suggesting that coinfection can substantially alter the inflammatory response compared to individual infections. However, interaction between cysticercosis and HIV did not show significant effect on the rest of the measured cytokines. The regression analysis results are further explained in S2 Text.

## Discussion

In our study, we explored the relationship between NCC and HIV, focusing on the peripheral immune response in coinfected patients. Our findings revealed a higher proportion of cysticercosis antigen positivity, which may highlight potential impact of HIV on cysticercosis infections on HIV+ people who are on effective ART. The fact that antibody positivity was equal between HIV+ and HIV- people speaks against this point. Another reason may be differences in infection pressure between the two groups or residual confounding for which we could not control for. There were no significant variations in cytokine levels between the different HIV and NCC co-infection groups. This observation indicates a potentially complex interplay between HIV and NCC that does not manifest in marked changes in cytokine profiles, even among those on effective ART. These findings challenge existing theories regarding their joint influence on the immune system.

Our findings are consistent with those of Schmidt et al. [39] and Serpa et al. [40] who found no significant differences in the presence of cysticercosis antibodies, or in the serological, clinical, and radiological presentation between HIV+ and HIV- patients coinfected with NCC. However, there was a notable disparity between antigen positivity and confirmed CT scan findings, indicating challenges in the diagnostic evaluation of NCC. This challenge is primarily due to the variable sensitivity and specificity of available diagnostic methods particularly in detecting cysts in certain brain regions such as the basal ganglia and brainstem, and/or in different cysticerci stages such as active NCC in subarachnoid space or intraventricular areas, where CT scans may be less accurate [41–44]. MRI offers superior sensitivity in identifying cysts within these intricate areas, although its availability and expertise are limited in most NCC endemic regions [41,43,45]. The disparity between antigen positivity and confirmed CT scan findings may also reflect the capacity of the antigen test used to detect viable cysts in parts of the body other that the brain [46]. This underscores the diagnostic challenge and highlights the need for comprehensive screening incorporating serological tests, imaging, and epidemiological assessments to identify and manage NCC [10,47].

Additionally, our findings of similar cytokine profiles between HIV+ and HIV- negative participants coinfected with cysticercosis, highlight the effectiveness of ART. This is evidenced by largely undetectable viral loads and reasonable CD4 counts in the HIV+ cohort, reflecting successful viral suppression and some restoration of immune function. This successful ART may also contribute to the masking of any additional immunological impacts of coinfection as it has been shown to reduce markers of systemic immune activation and latent infection [48,49]. Despite similar cytokine levels between these groups, HIV- participants with cysticercosis exhibited significantly depressed levels of IL-5, suggestive of a diminished eosinophil response critical for cyst clearance [50]. This indicates a potential strategic immune evasion and regulation by the cysticerci to favor its survival, supported by studies from Chavarria et al. [24] and Sáenz et al. [33] which noted specifically depressed cellular immune responses in symptomatic NCC patients and a depression in peripheral immune proliferation, particularly in symptomatic NCC. The absence of IL-5 depression in HIV+ individuals could be due to the complex interplay between HIV and the immune system. HIV infection is known to dysregulate the immune response, affecting various cytokines and immune cells [51].

The relationship between cytokine profiles and clinical presentations of NCC, particularly in terms of cyst activity, did not show significant differences across HIV statuses. This could suggest that the CNS environment may regulate immune cell activity and cytokine levels, preserving neural integrity despite infection [52]. Moreover, peripheral cytokine levels might not be reliable indicators of cyst activity or severity of NCC, suggesting that local immune responses within the CNS or at the site of cysts might be more indicative of disease activity. This is supported by the study by Carmen-Orozco et al. [53] which found that most of the pro-inflammatory-associated cytokines examined (IFN-γ, TNF-α, CSF-1, CSF-2, IL-1β, IL-1α) were increased in the area surrounding the viable cyst when compared to the non-infected tissue; however, the study did not find any significant changes in cytokine levels in brain tissue far from the cyst.

Despite our initial hypothesis, we did not observe any difference in Th1 and Th2 cytokines between patients with NCC and those without NCC for both HIV+ and HIV- groups. Similarly, there was no significant difference in cytokine levels when comparing active NCC to inactive NCC for most cytokines. However, for the HIV- group ICAM-1 levels exhibited a notable trend, with higher concentrations in patients with active NCC compared to those with inactive NCC (adjusted p = 0.0826). This finding suggests that active cysts may suppress certain immune responses, potentially as a strategic mechanism to evade host defenses [54]. Short-lived presentation of parasite-associated antigens by dendritic cells, leading to the suppression of immune responses has been reported as one the possible mechanisms for suppression [55]. This aligns with our observation of ICAM-1 trends, suggesting its potential role in indicating active infection. To highlight the complexity of immune response dynamics in NCC, Medina-Escutia et al. [56] found no evidence of immune suppression in patients with active NCC compared to healthy controls, while Nash et al. [57] reported increased inflammation associated with active NCC, underscoring the variable nature of the immune response in different contexts.

For the case of HIV+ group, the lack of statistically significant differences between the groups could suggest either antiretroviral therapy effectively restores immune responses or there exists an unexplored complexity in immune dynamics that we did not capture. Persistent immune activation in HIV+ individuals, even after achieving undetectable viral loads, has been reported and may contribute to the immune response we observed [58,59]. This warrants further investigation into the interplay between HIV and NCC.

Importantly, our study points to the potential influence of sex in modulating cytokine profiles in response to cysticercosis and HIV, with male sex being associated with decreased levels of IL-8, IL-17, IL-4, and ICAM-1, emphasizing a suppressive effect of male sex on these cytokines. This aligns with the findings from McClelland et al. [60] that reported increased inflammation in females compared to males in response to *T. solium* infection. Another study also found women showing increased levels of interleukins IL6, IL5, and IL10 in response to NCC [61]. This immunological difference means sex can influence, to some extent, the immune response to cysticercosis and disease severity, the number of cysts, the stage, and cyst location. This is also true for the context of HIV; female patients have been found to exhibit stronger alterations in the gut mucosal T-cell repertoire, with increased frequencies of Th1, Th17, and Th1/Th17-cell subsets compared to male patients [62]. While the exact mechanisms underlying these observations remain to be elucidated, it is plausible that hormonal differences and age-related changes in immune function could contribute to these effects. This sex-based disparity in the immune response to cysticercosis, as well as the modulating effects of HIV on this response, warrants further investigation to elucidate the underlying mechanisms.

Additionally, our study highlights the modulating effects of HIV on cytokine profiles, where being HIV positive was associated with an increase in several cytokines, including TNF-

α, IL-8, IFN-γ, IL-17, IL-12, IL-6, and IL-5, indicating potent stimulatory effects of HIV infection. However, contrary to initial hypotheses, the coexistence of HIV with cysticercosis did not lead to significant alterations in the levels of TNF-α, IFN-γ, IL-5, and VCAM-1. This suggests that the inflammatory response in coinfected individuals does not differ substantially from that observed in HIV infection alone, highlighting the dominant influence of HIV over the immune evasion mechanisms potentially employed by the *T. solium* parasite. [63]. This is in contrast to HIV and tuberculosis coinfections, where an additive effect on cytokine levels is typically observed [64]. These findings emphasize the unique immunological interactions in cysticercosis coinfection and underscore the need for tailored therapeutic approaches that specifically address the complex dynamics of immune response alterations driven by HIV and other coinfecting pathogens.

This study addresses a critical gap in the literature concerning the peripheral immune responses to HIV and NCC coinfection. Moreover, the cytokine modulation observed offers insights into potential immunotherapeutic targets, particularly in enhancing disease management strategies for coinfected individuals. To extend this research, future studies should focus on larger, more diverse cohorts to validate our findings and explore the mechanistic bases of these immune interactions. Longitudinal studies could also provide deeper insights into how these cytokine profiles evolve with disease progression and treatment, potentially guiding more effective clinical interventions.

## Limitations

While our study provides critical insights, it is not without limitations. The goodness of fit analyses yielded only a poor fit of the models. In addition, the lack of full-body CT scans might have missed cysts outside the brain, and the preference for CT over MRI could underestimate active cyst prevalence. Furthermore, the absence of ART-naïve HIV+ patients in our sample may limit the generalizability of our findings. Also, the sample size may not have been large enough to detect differences given our complex study design. Addressing these limitations in future research, perhaps through methodological enhancements or by incorporating advanced immunological assays, will be crucial in refining our understanding of these complex interactions.

## Conclusion

Our findings suggest that coinfection with HIV and neurocysticercosis (NCC) does not significantly modulate peripheral cytokine profiles, despite the elevated IL-5 levels observed in HIV-positive individuals with cysticercosis. While this observation hints at potential immunological interplay, the lack of statistical significance in broader cytokine modulation warrants cautious interpretation. Notably, the higher cysticercosis antigen positivity among HIV-positive participants raises biological implications for NCC progression in the context of HIV coinfection. However, our results underscore the need for robust, statistically significant data to draw conclusive statements about the immunological impacts of coinfection. Future studies with larger sample sizes and advanced statistical analyses are warranted to further elucidate these interactions and inform the understanding and management of coinfected patients.

## Supporting information

**S1 Text. This document includes the baseline demographic and clinical characteristics of the study participants.**
(DOCX)

**S2 Text. This document provides detailed cytokine profile data and results of the multivariable regression analysis, highlighting the associations between HIV status, cysticercosis status, and cytokine levels.**
(DOCX)

**S1 Table. This Excel file contains the data tables supporting Fig 1 to 4, including median concentrations ± SEM for all cytokines evaluated.**
(XLSX)

**S1 Data. This Excel file contains all the data used in the analyses described in this manuscript.** The second worksheet includes metadata describing all variables and units.
(XLSX)

## Author Contributions

**Conceptualization:** Sayoki Mfinanga, Veronika Schmidt, Hélène Carabin, Andrea Sylvia Winkler, Bernard J. Ngowi, Clarissa Prazeres da Costa.

**Data curation:** Yakobo Leonard Lema, Ulrich Fabien Prodjinotho, Charles Makasi, Eligius F. Lyamuya, Mkunde Chachage, Clarissa Prazeres da Costa.

**Formal analysis:** Yakobo Leonard Lema, Ulrich Fabien Prodjinotho, Mary-Winnie A. Nanyaro, Sayoki Mfinanga, Dominik Stelzle, Hélène Carabin, Eligius F. Lyamuya, Mkunde Chachage, Clarissa Prazeres da Costa.

**Funding acquisition:** Sayoki Mfinanga, Veronika Schmidt, Hélène Carabin, Andrea Sylvia Winkler, Bernard J. Ngowi, Clarissa Prazeres da Costa.

**Investigation:** Yakobo Leonard Lema, Ulrich Fabien Prodjinotho, Charles Makasi, Mary-Winnie A. Nanyaro, Andrew Martin Kilale, Sayoki Mfinanga, Dominik Stelzle, Veronika Schmidt, Andrea Sylvia Winkler, Bernard J. Ngowi, Clarissa Prazeres da Costa.

**Methodology:** Yakobo Leonard Lema, Ulrich Fabien Prodjinotho, Charles Makasi, Mary-Winnie A. Nanyaro, Sayoki Mfinanga, Dominik Stelzle, Veronika Schmidt, Hélène Carabin, Andrea Sylvia Winkler, Bernard J. Ngowi, Clarissa Prazeres da Costa.

**Project administration:** Yakobo Leonard Lema, Ulrich Fabien Prodjinotho, Mary-Winnie A. Nanyaro, Andrew Martin Kilale, Sayoki Mfinanga, Dominik Stelzle, Andrea Sylvia Winkler, Bernard J. Ngowi, Clarissa Prazeres da Costa.

**Resources:** Mkunde Chachage, Clarissa Prazeres da Costa.

**Supervision:** Andrew Martin Kilale, Sayoki Mfinanga, Andrea Sylvia Winkler, Eligius F. Lyamuya, Bernard J. Ngowi, Mkunde Chachage, Clarissa Prazeres da Costa.

**Validation:** Yakobo Leonard Lema, Ulrich Fabien Prodjinotho, Andrea Sylvia Winkler, Eligius F. Lyamuya, Bernard J. Ngowi, Mkunde Chachage, Clarissa Prazeres da Costa.

**Visualization:** Yakobo Leonard Lema, Ulrich Fabien Prodjinotho, Dominik Stelzle, Eligius F. Lyamuya, Mkunde Chachage, Clarissa Prazeres da Costa.

**Writing – original draft:** Yakobo Leonard Lema, Ulrich Fabien Prodjinotho, Charles Makasi, Andrea Sylvia Winkler, Eligius F. Lyamuya, Bernard J. Ngowi, Mkunde Chachage, Clarissa Prazeres da Costa.

**Writing – review & editing:** Yakobo Leonard Lema, Ulrich Fabien Prodjinotho, Charles Makasi, Mary-Winnie A. Nanyaro, Andrew Martin Kilale, Sayoki Mfinanga, Dominik

Stelzle, Veronika Schmidt, Hélène Carabin, Andrea Sylvia Winkler, Eligius F. Lyamuya, Bernard J. Ngowi, Mkunde Chachage, Clarissa Prazeres da Costa.

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
