## [Decision Letter · Decision Letter 0]

31 Jan 2024

Dear Dr Lema,

Thank you very much for submitting your manuscript "Evaluating the modulation of peripheral immune profile in People Living with HIV and Neurocysticercosis" for consideration at PLOS Neglected Tropical Diseases. As with all papers reviewed by the journal, your manuscript was reviewed by members of the editorial board and by several independent reviewers. In light of the reviews (below this email), we would like to invite the resubmission of a significantly-revised version that takes into account the reviewers' comments. 

We cannot make any decision about publication until we have seen the revised manuscript and your response to the reviewers' comments. Your revised manuscript is also likely to be sent to reviewers for further evaluation.

Sincerely,

Gary L. Simon

Academic Editor

Uriel Koziol

Section Editor

Reviewer's Responses to Questions

**Key Review Criteria Required for Acceptance?**

**Methods**

-Are the objectives of the study clearly articulated with a clear testable hypothesis stated?

-Is the study design appropriate to address the stated objectives?

-Is the population clearly described and appropriate for the hypothesis being tested?

-Is the sample size sufficient to ensure adequate power to address the hypothesis being tested?

-Were correct statistical analysis used to support conclusions?

-Are there concerns about ethical or regulatory requirements being met?

Reviewer #1: The description of the study groups is not clear; the text, in general, needs to be improved.

In the methodology section, it is not explained why they decided to evaluate the presence of eggs or proglottids in feces (taeniasis).

Reviewer #2: Objectives are clear and methodology is also well defined. Ethical permission is taken from institute and population is also clearly described.

Reviewer #3: Sample size may be too small. Many findings appear to be trends but not statistically significant.

**Results**

-Does the analysis presented match the analysis plan?

-Are the results clearly and completely presented?

-Are the figures (Tables, Images) of sufficient quality for clarity?

Reviewer #1: The section needs improvement in terms of presentation of results (e.g.The paragraph from line 231 to 235 is a summary of the previous paragraph (218-230), the narrative is in many sections contradictory.

Reviewer #2: Results need to be presented in more clear way as by this plot their is some confusion in presentation of data. or the point authors are making are not visible.

Reviewer #3: (No Response)

**Conclusions**

-Are the conclusions supported by the data presented?

-Are the limitations of analysis clearly described?

-Do the authors discuss how these data can be helpful to advance our understanding of the topic under study?

-Is public health relevance addressed?

Reviewer #1: The study is important and has considerable value due to the type of samples used and groups analyzed; however, it is necessary to rewrite the text, standardize styles and ways of presenting the study.

Reviewer #2: Majorly supported.

Reviewer #3: (No Response)

**Editorial and Data Presentation Modifications?**

Reviewer #1: Line 61-62: “Neurocysticercosis is caused by the larval stage of tapeworm Taenia solium, and is associated with seizures and other..”

Line 64: Indicate which regions

Line 78: “The ingestion of eggs shed in the stool of a human tapeworm carrier leads to the lodging of Taenia solium larvae in the central nervous system, causing NCC”. This expression is not entirely correct, parasites can establish and develop in muscles, connective tissue, eyes, and brain.

Line: 80: Change invade to penetrate

Line 94: Which cells?

Line 129: Reference?

Reviewer #2: (No Response)

Reviewer #3: The discussion section is far too long, in part because the authors comment on numerous small differences among groups rather than focusing on those that are well-supported by the data, i.e. those that are statistically significant.

**Summary and General Comments**

Reviewer #1: It is necessary for the authors to rewrite the text before publication, it needs considerable improvement.

Reviewer #2: (No Response)

Reviewer #3: The major issue with this manuscript is that the conclusions are overstated. Differences among groups that are not statistically significant should not be presented as conclusions. The actual p-values should be included in figures where the authors claim there is a difference and the discussion should help readers understand the limitations of the sample sizes and effect sizes that contribute to the uncertainty in the findings.

PLOS authors have the option to publish the peer review history of their article (what does this mean?). If published, this will include your full peer review and any attached files.

Reviewer #1: No

Reviewer #2: Yes: Amit Prasad

Reviewer #3: No
---

## [Editor Report · Decision Letter 1]

7 Jun 2024

Dear Dr Lema,

Thank you very much for submitting your manuscript "Evaluating the modulation of peripheral immune profile in People Living with HIV and (Neuro)cysticercosis" for consideration at PLOS Neglected Tropical Diseases. As with all papers reviewed by the journal, your manuscript was reviewed by members of the editorial board and by several independent reviewers. The reviewers appreciated the attention to an important topic. Based on the reviews, we are likely to accept this manuscript for publication, providing that you modify the manuscript according to the review recommendations. 

This paper still needs further revision. Tables 1 and 3 provide no useful information that could not be included in few lines of text, and could be transferred to the supplementary material. Similarly, the figures would be better expressed in the text with standard deviations. More important is that the authors now recognize the lack of statistical significance, but then suggest that the trends are suggestive of changes that might become apparent with a larger sample size. Unless the p-value for trends lies between 0.05 and 0.1 it is difficult to support the concept that the trend could become clinically significant.

Sincerely,

Gary L. Simon

Academic Editor

Uriel Koziol

Section Editor

This paper still needs further revision. Tables 1 and 3 provide no useful information that could not be included in few lines of text. Similarly, the figures would be better expressed in the text with standard deviations. More important is that the authors now recognize the lack of statistical significance, but then suggest that the trends are suggestive of changes that might become apparent with a larger sample size. Unless the p-value for trends lies between 0.05 and 0.1 it is difficult to support the concept that the trend could become clinically significant.

Figure Files:

Data Requirements:

Reproducibility:

References

---

## [Editor Report · Decision Letter 2]

5 Jul 2024

Dear Dr Lema,

We are pleased to inform you that your manuscript 'Evaluating the modulation of peripheral immune profile in People Living with HIV and (Neuro)cysticercosis' has been provisionally accepted for publication in PLOS Neglected Tropical Diseases.

Best regards,

Gary L. Simon

Academic Editor

Uriel Koziol

Section Editor

---

## [Editor Report · Acceptance letter]

26 Jul 2024

Dear Dr Lema,

We are delighted to inform you that your manuscript, "Evaluating the modulation of peripheral immune profile in People Living with HIV and (Neuro)cysticercosis," has been formally accepted for publication in PLOS Neglected Tropical Diseases.

Best regards,

Shaden Kamhawi

co-Editor-in-Chief

Paul Brindley

co-Editor-in-Chief
